# The effectiveness of low-dosed outpatient biopsychosocial interventions compared to active physical interventions on pain and disability in adults with nonspecific chronic low back pain: A protocol for a systematic review with meta-analysis

**Martin Hochheim**[1,2]*, **Philipp Ramm**[2], **Volker Amelung**[1]

**1** Institute of Epidemiology, Social Medicine and Health System Research, Medizinische Hochschule Hannover (MHH), Hannover, Germany, **2** Generali Health Solutions GmbH (GHS), Köln, Germany

* Martin.J.Hochheim@stud.mh-hannover.de

## Abstract

### Background

Best outpatient treatment of nonspecific chronic low back pain is high-dosed exercise that is maintained after therapy. Primary care biopsychosocial intervention (PCBI) is an outpatient multicomponent intervention that includes an active component (exercise, physical activity, or physiotherapy) and at least one psychological, social, or occupational component. Research has shown that PCBI can reduce pain intensity and disability. While scattered studies support low-dosed (<15 treatment hours) PCBI, there is no systematic review comparing the effectiveness of low-dosed PCBI treatment with traditional physical activity interventions in adults with nonspecific chronic low back pain (CLBP).

### Inclusion criteria

Randomised controlled trials that evaluate low-dosed outpatient biopsychosocial interventions compared to physical treatment with an active component such as exercise, physical activity or usual physiotherapy treatment for adult participants (18 years or older) who suffer from nonspecific CLBP will be included.

### Methods

A comprehensive search of multiple databases will be used to find relevant studies. The databases will be searched from inception to December 2021, with English or German language restrictions imposed. Keywords and derivatives of "chronic back pain", "exercise intervention", "cognitive-behavioral therapy", "primary care" and "randomized controlled trials" will be used. Sources will include CINAHL, Cochrane Central Register of Controlled

**Data Availability Statement:** No datasets were generated or analysed during the current study. All

relevant data from this study will be made available upon study completion.

**Funding:** The authors received no specific funding for this work.

**Competing interests:** The authors have declared that no competing interests exist.

**Abbreviations:** *Abbreviation*, *Explanation*; CLBP, Nonspecific chronic low back pain; GRADE, Grading of Recommendations, Assessment, Development and Evaluation; HRQoL, Health-related quality of life; LBP, Low back pain; MBR, Multidisciplinary biopsychosocial rehabilitation; NRS, Numerical rating scale; ODI, Oswestry Disability Index; PCBI, Primary care biopsychosocial intervention; PRISMA, Preferred Reporting Items for Systematic Reviews and Meta-analyses; RCT, Randomised controlled trials; RMDQ, Roland-Morris Disability Questionnaire; SF-12, Short Form 12; SF-36, Short Form 36; SMD, Standardised mean differences; SoF, Summary of Findings; VAS, Visual analogue scales.

Trials (CENTRAL), Ovid Medline, Physiotherapy Evidence Database (PEDro), PubMed and Web of Science.

## Discussion

To our knowledge, this will be the first systematic review and meta-analysis of narrowly defined low- dosed PCBI across populations with nonspecific chronic low back pain. The objective of this review is to evaluate the effectiveness of low-dosed outpatient biopsychosocial interventions versus physical active interventions on pain intensity and disability in adults with CLBP. This study will provide evidence that could improve treatment options for patients with nonspecific CLBP.

## Trail registration

**Systematic review registration number:** PROSPERO 2022 CRD42022302771. https://www.crd.york.ac.uk/prospero/display_record.php?ID=CRD42022302771

## Introduction

Globally, nonspecific low back pain (LBP) is one of the leading causes of years of disability and absenteeism from work [1–3]. In 2017, an estimate of 577 million people around the world suffered from LBP. The lifetime prevalence of LBP is 70 to 80% [4–6]. Most of those affected recover within the first six weeks. However, 10 to 40% remain in a state of nonspecific chronic low back pain (CLBP) [5,6].

CLBP is defined as pain below the costal margin and the gluteal region, with or without radiation to one or both legs, that persists for 12 weeks or more [7]. CLBP, as a major health problem [4], is a long-lasting, extremely common musculoskeletal condition without a clear pathology. It limits activities and results in a high personal, social, and economic burden [8,9]. Patients with CLBP have an increased presence of psychological factors such as kinesiophobia, maladaptive coping, anxiety, depression, catastrophising, or low self-efficacy [10]. Consequently, CLBP is often conceptualised as a biopsychosocial condition, i.e. as a complex and dynamic interaction between physical, psychological, and social elements [11]. Focusing on the different influencing factors of CLBP inpatient multidisciplinary biopsychosocial rehabilitation (MBR), an intensive, multicomponent inpatient intervention is frequently recommended in clinical treatment guidelines for CLBP [12]. As the condition is already chronic, the primary objective of MBR is to restore the daily functioning of the participant, not to cure the pain [13]. It is usually provided by multidisciplinary teams of healthcare professionals in rehabilitation centres or specialised pain clinics and has been shown to be very effective in reducing pain and disability [14,15]. Outcomes are generally patient-reported physical functioning and pain intensity in the short, medium, and long term.

However, MBR in secondary or tertiary care is limited in numbers, not widely available in healthcare settings, is associated with long waiting times, and requires a large number of treatment hours from different healthcare professionals [11,16,17]. This leads to a high economic burden on individuals and the health system [18]. According to habit theory, individuals changing their context are more likely to perform non-habitual behaviour. But as they return to their home environment, a relapse to old habits is more likely to occur [19].

The best evidence for effective outpatient treatment of CLBP is exercise delivered in a high-dose of at least 20 treatment hours that is maintained continuously after the initial intervention [20,21]. It should be noted that effective exercise therapy, which reduces pain and disability, often does not correlate with improved physical function of the musculoskeletal system [22]. It may be that other exercise-induced changes such as emotional functions and improved cognitions (e.g., reduced anxiety, improved self-efficacy, less catastrophizing and less fear-avoidance behaviour) influence pain and disability more than changes in muscle strength, mobility, or muscular endurance [21].

To increase the chance of long-term training maintenance and improved psychological outcomes, a focus could be placed on participants' motivation, self-efficacy, and change in daily habits with a combined approach of exercise and psychological therapy in the form of a primary care biopsychosocial intervention (PCBI). Furthermore, outpatient PCBI could support people in establishing plans to cope with difficult situations, anticipate and practice high-risk situations in the social or physical environment [23]. Another advantage of PCBI is its possible administration by trained physical therapists [18]. Therefore, it seems to be an interesting and potentially cost-effective approach, as a physiotherapist session in primary care is more accessible and cheaper than multidisciplinary treatment [17]. If it were further possible to reduce the dosages of outpatient active treatment sessions by using additional psychosocial elements and still receive positive outcomes, the treatment cost could possibly be further reduced. Thus, treatment could be offered to a larger population, with a shorter waiting time.

On the effectiveness of general PCBI compared to active treatment, there are mixed results with conflicting evidence. Some studies indicate a beneficial effect in favour of PCBI [24,25]. While others report no difference between PCBI and conservative activity treatments in terms of effectiveness [26,27]. Positive results of low-dose PCBI have been achieved with cognitive functional therapy [28–30].

The last systematic review on PCBI covered the scientific literature until 2015 and identified promising effects [16]. A systematic review with meta-analysis, which covered group-based physiotherapy-led behavioral psychological interventions identified a small but significant long-term pain reduction compared to other active treatments [31]. As it covered only group-based interventions and did not differ in dose, the results can only indicate that outpatient treatment is a promising alternative to other active treatments. The third systematic review in this area of research focused on cognitive-behavioural based interventions delivered by a physiotherapist. The authors found high-quality evidence for reduced disability compared to no treatment, usual care, or other active treatment. However, they also had a broad focus and included all types of controls, making a direct comparison difficult [17].

A preliminary search of PROSPERO, MEDLINE, the Cochrane Database of Systematic Reviews and the JBI Evidence Synthesis was conducted and no current or underway systematic reviews have focused on low dosages of PCBI compared to conservative exercise therapy. Therefore, the objective of this review is to understand whether a low-dose (and therefore less costly) PCBI with a maximum of 15 treatment hours is more effective than exercise treatment alone for patients with nonspecific CLBP. Special interest is in long-term outcomes (>1 year).

## Review question

Is a low-dosed primary care biopsychosocial intervention (PCBI), consisting of an active physical component and at least one psychological, social, or occupational component, with a maximum of 15 treatment hours more effective in reducing pain intensity and improving physical function than other active outpatient physical treatment approaches for adult patients with nonspecific CLBP?

## Inclusion criteria

### Participants

Studies with adult participants (18 years or older) who suffer from nonspecific chronic low back pain are included. Nonspecific chronic low back pain is defined as pain below the costal margin and the gluteal region, with or without radiation to one or both legs, that persists for 12 weeks or more.

Trials are excluded when the sample includes participants with acute or subacute LBP (unless subacute LBP subjects comprised 15% or less of the total study population (≥85% should be CLBP), or results of patients with CLBP are presented separately), participants with specific low back pain due to a known, specific pathology (i.e., stenosis, spondyloarthritis, ankylosing, fractures, infection, or spinal cord compression) or women suffering from pregnancy-related back pain.

**Intervention(s).**   This review will consider studies that evaluate low-dosed outpatient biopsychosocial interventions. Following previous systematic reviews, biopsychosocial is defined as a multicomponent intervention that includes an active component (exercise, physical activity, or physiotherapy) and at least one psychological, social, or occupational component [16,32,33]. As many different types of exercise have shown to be beneficial for patients with CLBP, there is no clear consensus on a specific type of exercise [20]. Therefore, we will include all different types of exercises. They must involve specific activities, postures, or movements with the aim of improving physical health in chronic low back pain patients [20]. Among others, this can include muscle or core strengthening, flexibility and mobilisation exercises, aerobic exercises, Pilates, Yoga or other specific exercises.

A low dosage is defined as a maximum of 15 face-to-face treatment hours. Unsupervised home exercise does not count toward treatment hours. Included are individually (one-to-one) and group-supervised (two or more participants) therapy sessions.

The active component must be delivered face-to-face. Interventions must be delivered in primary care (referral by general practitioner, physiotherapist in local facilities, primary care practice in a hospital, other outpatient healthcare professions). Mono- and multidisciplinary delivered interventions will be included.

The psychological, social, or occupational component(s) can have other types of delivery (telephone, web-based). These components aim to improve (knowledge of) cognitive, emotional, social, or lifestyle factors or coping responses [29,33,34]. Examples are cognitive behavioral therapy (CBT), acceptance and commitment therapy, pain neuroscience education (PNE), motivation or self-regulation training.

According to the CLBP treatment guidelines, first-line interventions that are not recommended and feature only passive therapies (e.g., acupuncture, electrotherapy, traction, massage), spinal surgery, or sole pharmacological treatment will be excluded [12,35]. Studies with completely independent, online, or telephone-based interventions (all components not equal to face-to-face), or interventions delivered inpatient (e.g., in pain clinics or rehabilitation centres) will be excluded. Studies in an occupational setting will also be excluded.

**Comparator(s).**   This review will consider studies that compare low-dosed (<15h) outpatient biopsychosocial interventions with physical treatment with an active component such as exercise, physical activity or usual physiotherapy treatment.

The comparison can be delivered in any type of delivery mode (face-to-face, as home training, web-based, app, etc.). Excluded are control groups that are formed from a waiting-list, undergo a complete biopsychosocial intervention (as defined above), get invasive therapy, or receive a sole pharmacological intervention.

**Outcomes.** This review will consider studies that include clinical outcomes of physical functioning/disability and pain intensity.

According to the recommendation for core outcomes in clinical trials with participants with nonspecific low back pain [9], the outcomes of interest are physical functioning (measured with the Oswestry Disability Index (ODI) version 2.1a [36,37] or the Roland-Morris Disability Questionnaire (RMDQ) of 24-items [38]) and pain intensity (measured with the numerical rating scale (NRS) or visual analogue scales (VAS) [39]).

Outcomes of secondary interest are HRQoL measured with SF-12 [40], SF-36 [41] (dimensions: physical function, general health, mental health) or 10-item PROMIS Global Health short form [42]) and all adverse events.

## Types of studies

We will only include randomised controlled trials (RCTs) published in full text in peer reviewed journals. We include trials from the journals' inception time to December 31, 2021 that were published in English or German language and enrolled adults with CLBP in a biopsychosocial intervention.

## Methods

The proposed systematic review will be conducted in accordance with the JBI methodology for systematic reviews of effectiveness evidence [43]. According to the guidelines, our systematic review protocol was registered with the International Prospective Register of Systematic Reviews (PROSPERO 2022) on 27 February 2022 (registration number CRD42022302771). We followed the Preferred Reporting Items for Systematic review and Meta-Analyses Protocols (PRISMA-P) [44] guideline and the available checklist (see S2 Appendix).

## Search strategy

The search strategy will aim to locate published studies. In this review, a three-step search strategy will be used. First, an initial limited search of MEDLINE (Ovid) and CINAHL (EBSCO) will be undertaken to identify articles on the topic. The text words contained in the titles and abstracts of relevant articles, and the index terms used to describe the articles, will be used to develop a full search strategy for:

- CINAHL.

- Cochrane Central Register of Controlled Trials (CENTRAL).

- Ovid Medline.

- Physiotherapy Evidence Database (PEDro).

- PubMed.

- Web of Science.

The databases will be additionally searched for relevant systematic reviews and meta-analyses. Titles, abstracts, key words, and reference lists will be scanned to refine search terms.

The search strategy, including all identified keywords and index terms, will be adapted for each included database and/or information source (see S1 Appendix). The reference list of all included sources of evidence will be screened for additional studies. A forward reference search of all included studies will be performed using the R package *citationchaser* [45]. Studies

published in English or German will be included. Studies published since their inception date will be included.

## Study selection

Following the search, all identified citations will be collated and uploaded to Mendeley Desktop 1.19.8/2020 (Mendeley Ltd., London, UK) and duplicates will be removed. Following a pilot test, titles and abstracts will then be screened by two independent reviewers (MH and PR) for assessment against the inclusion and exclusion criteria of the review in PICO Portal [46]. Potentially relevant studies will be retrieved in full and their citation details will be imported into the PICO Portal. The full text of the selected citations will be assessed in detail against the inclusion and exclusion criteria by two independent reviewers (MH and PR). Reasons for exclusion of full text articles that do not meet the inclusion criteria will be recorded and reported in the systematic review. Any disagreements that arise between the reviewers at each stage of the selection process will be resolved through discussion, or with at least one additional reviewer. The results of the search and the study inclusion process will be fully reported in the final systematic review and presented in a Preferred Reporting Items for Systematic Reviews and Meta-analyses (PRISMA) flow diagram [47].

## Assessment of methodological quality

Eligible studies will be critically appraised by two independent reviewers (MH and PR) at the study level using standardised critical appraisal instruments from JBI for experimental studies. The authors of papers will be contacted to request missing or additional data for clarification, where required. Any disagreements that arise will be resolved through discussion, or with a third reviewer. The results of critical appraisal will be reported in narrative form and in a table.

Following critical appraisal, studies that do not meet a certain quality threshold will be excluded. This decision will be based on the JBI 13-item checklist for randomised controlled trials [48]. Studies will be excluded if one or more of the following checklist questions is answered with no:

- Was true randomization used for assignment of participants to treatment groups? (Question 1 / selection bias)

- Were treatment groups similar at baseline? (Question 3 / selection bias)

- Were treatment groups treated identically other than the intervention of interest? (Question 7 /performance bias)

- Were outcomes measured in the same way for treatment groups? (Question 10 / measurement bias)

The methodological quality will be assessed into the three categories: Low risk of bias ($> =$ 9 questions answered with yes), some concerns ($> = 6$ questions answered with yes) or high risk of bias ($<6$ questions answered with yes).

## Data extraction

Data will be extracted from studies included in the review by an independent reviewer (MH) using the data extraction module of the PICO Portal [46]. The data extracted will be checked by a second review author (PR). Any disagreements that arise between the reviewers will be resolved through discussion or with a third reviewer.

The data extracted will include specific details on (1) study characteristics (number of participants, age, sex, length of follow-up), (2) type of interventions (delivery, dosage, content), (3) baseline and follow-up patient-reported outcome measures of significance to assess pain intensity and disability, and (4) summary of findings. In case there are multiple publications regarding one RCT, all available publications will be checked and relevant data extracted. To avoid double counting that would bias the meta-analysis, the outcome measure data for eligible interventions will be selected based on the follow-up time. In case of reporting the same outcomes of the same RCT in different reports, the report with the higher methodological quality will be selected. The authors of papers will be contacted to request missing or additional data, when required. In cases where authors are uncontactable, a web-based tool will be used to extract raw data from the graphs [49]. Any disagreements that arise between the reviewers will be resolved through discussion or with a third reviewer.

## Data synthesis

A descriptive synthesis of outpatient biopsychosocial interventions with active treatment as a comparator will be performed. Studies will, where possible, be pooled in a statistical meta-analysis using RevMan Web from the Cochrane Collaboration [50]. Effect sizes will be expressed as standardised mean differences (SMD), and their 95% confidence intervals will be calculated for analysis. Statistical analyses will be performed with a random effects model using the *DerSimonian and Laird method* [51]. This model not only accounts for the statistical heterogeneity of the included studies, but allows for generalisation beyond these studies [52,53].

Subgroup analyses will be performed when there is sufficient data to investigate different types (individual versus group setting) or length of follow-up (short, medium, long-term) of biopsychosocial interventions. After completion of the review and based on the findings of the included studies, we may perform additional subgroup analyses if they are believed to contribute valuable information.

Following the meta-analysis, a sensitivity analysis will be performed to assess the impact of risk of bias on the results. In the sensitivity analysis, studies classified as having a high risk of bias will be omitted.

Heterogeneity between the studies will be evaluated graphically with forest plots and funnel plots (if there are 10 or more studies included in the meta-analysis) and statistically with the chi-squared and I squared tests. As the effect sizes will be expressed in SMDs, statistical tests for funnel plot asymmetry will not be performed [54]. The level of heterogeneity will be interpreted according to the latest version of the Cochrane Handbook for Systematic Reviews of Interventions, which provides the following guidance for the I2 statistic: '0% to 40%: might not be important; 30% to 60%: may represent moderate heterogeneity; 50% to 90%: may represent substantial heterogeneity; and 75% to 100%: considerable heterogeneity [53].

Should meta-analysis due to clinical and methodological heterogeneity not be possible, a narrative synthesis will be performed. The findings will then be presented in narrative form, including tables and figures to aid in data presentation, where appropriate.

## Assessing certainty in the findings

The Grading of Recommendations, Assessment, Development and Evaluation (GRADE) approach for grading the certainty of evidence will be followed [55] and a Summary of Findings (SoF) will be created using GRADEpro GDT (McMaster University and Evidence Prime, ON, Canada) [56]. This will be carried out by two independent reviewers (MH and PR) at the outcome level. Any disagreements that arise between the reviewers will be resolved through

discussion or with a third reviewer. The authors of papers will be contacted to request missing or additional data for clarification, where required.

The SoF will present the following information, where appropriate: Patients or population, setting, intervention, comparison, number of participants, anticipated absolute effects for treatment and control, and a ranking of the quality of the evidence based on the risk of bias, directness, heterogeneity, precision, and risk of publication bias of the review results. The outcomes reported in the SoF will be pain intensity and functional limitations.

## Discussion

To our knowledge, this will be the first systematic review and meta-analysis of narrowly defined low-dosed PCBI in populations with nonspecific chronic low back pain. Due to the prevalence of CLBP throughout the world and its huge economic burden on individuals and health systems, we believe that this research will be interesting for decision makers worldwide. Also, an impact on treatment decisions is expected.

With the focus on long-term outcomes, we take into account that low back pain symptoms show a similar pattern of improvement in the short term in a wide range of interventions [57]. Using not only pain intensity, but also disability and HRQoL as outcomes of interest, we further take the nature of CLBP as a chronic condition into account. Our aim is to evaluate whether the outcomes can be long-lastingly improved to restore daily functioning as much as possible. -In doing this, our aim is to assess whether low-dosed PCBI can improve CLBP treatment outcomes. Depending on the results, the treatment and access options of further health programmes could be improved for a wide audience. If low-dosed biopsychosocial interventions are more effective than active treatments alone, our objective is to further identify which combination of biopsychosocial ingredients delivers the best outcomes to make effective treatment procedures accessible to a broad population at reduced costs due to minimised treatment time.

Due to the worldwide burden of CLBP, we expect the results of this systematic review to be of considerable interest to health care professionals, academics, guideline developers, and decision makers. We will widely distribute the findings through academic publications, conference presentations, and communication with healthcare providers.

## Supporting information

**S1 Appendix. Literature search strategy.**
(DOCX)

**S2 Appendix. PRISMA-P checklist.**
(DOCX)

## Acknowledgments

This systematic review is to count towards the dissertation of MH through Hannover Medical School (MHH).

## Author Contributions

**Conceptualization:** Martin Hochheim, Philipp Ramm, Volker Amelung.

**Investigation:** Martin Hochheim.

**Methodology:** Martin Hochheim.

**Project administration:** Martin Hochheim.

**Visualization:** Volker Amelung.

**Writing – original draft:** Martin Hochheim.

**Writing – review & editing:** Martin Hochheim, Philipp Ramm.

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
