## [Decision Letter · Decision Letter 0]

4 Aug 2022

PONE-D-22-15969The effectiveness of low-dosed outpatient biopsychosocial interventions compared to active physical interventions on pain and disability in adults with nonspecific chronic low back pain: a protocol for a systematic review with meta-analysis

PLOS ONE

Dear Dr. Hochheim,

Thank you for submitting your manuscript to PLOS ONE. After careful consideration, we feel that it has merit but does not fully meet PLOS ONE’s publication criteria as it currently stands. Therefore, we invite you to submit a revised version of the manuscript that addresses the points raised during the review process.

Both reviewers has now completed their suggestions. Please find these comments below. Your paper should be revised and resubmitted.

We look forward to receiving your revised manuscript.

Kind regards,

Fatih Özden, PhD

Academic Editor

PLOS ONE

Journal Requirements:

Reviewers' comments:

Reviewer's Responses to Questions

**Comments to the Author**

1. Does the manuscript provide a valid rationale for the proposed study, with clearly identified and justified research questions?

Reviewer #1: Yes

Reviewer #2: Yes

2. Is the protocol technically sound and planned in a manner that will lead to a meaningful outcome and allow testing the stated hypotheses?

Reviewer #1: Partly

Reviewer #2: Partly

3. Is the methodology feasible and described in sufficient detail to allow the work to be replicable?

Reviewer #1: Yes

Reviewer #2: Yes

4. Have the authors described where all data underlying the findings will be made available when the study is complete?

Reviewer #1: Yes

Reviewer #2: Yes

5. Is the manuscript presented in an intelligible fashion and written in standard English?

Reviewer #1: Yes

Reviewer #2: Yes

6. Review Comments to the Author

You may also provide optional suggestions and comments to authors that they might find helpful in planning their study.

Reviewer #1: This study protocol aimed to investigate whether a low-dose (and therefore less costly) PCBI with a maximum of 15 treatment hours is more effective than exercise treatment alone for patients with nonspecific CLBP. The strength of this study was to conduct analysis using a rigorous method of systematic reviews. However, there were some concerns in this study.

First, in the Abstract and the Types of studies, the authors had better revise the search date to July 2022. I consider the latest search date is better for a comprehensive search.

Second, in the Abstract and the Search strategy, they had better add Embase as one of the databases they will search. Cochrane Handbook for Systematic Reviews of Interventions Version 6.3, 2022 has recommended the search for CENTRAL, MEDLINE, and Embase in Cochrane reviews in the section 4.3.1.1 Introduction to bibliographic databases (Higgins JPT, Thomas J, Chandler J, Cumpston M, Li T, Page MJ, Welch VA (editors). Cochrane Handbook for Systematic Reviews of Interventions version 6.3 (updated February 2022). Cochrane, 2022. Available from www.training.cochrane.org/handbook.).

Third, in the Abstract and the Search strategy, they had better add ClinicalTrials.gov and WHO ICTRP as the databases they will search. These databases are important to search ongoing studies.

Fourth, they had better add “all adverse events” to outcomes.

Fifth, they had better revise the search strategies. I consider that the standard search strategy for systematic reviews is “Participants” AND “Interventions” AND the RCT filter. The standard search strategy includes two “AND”. However, the search strategy in S1 Appendix includes four “AND” in #34 in Medline (Ovid). The search strategy may have the possibility to miss eligible studies by using too many “AND”. Therefore, I recommend they consult an information specialist, a search coordinator, or a librarian to develop search strategies for all databases to conduct a comprehensive search.

Reviewer #2: The age range considered is from 18 years where chronic LBP might not be relevant also the exercises program are not defined well.

7. PLOS authors have the option to publish the peer review history of their article (what does this mean?). If published, this will include your full peer review and any attached files.

Reviewer #1: **Yes: **Masahiro Banno, MD, PhD

Reviewer #2: **Yes: **Dr. PREM KUMAR BHOJARA

---

## [Author Response · Author response to Decision Letter 0]

14 Aug 2022

1. First, in the Abstract and the Types of studies, the authors had better revise the search date to July 2022. I consider the latest search date is better for a comprehensive search.

Dear Dr. Banno,

Thank you very much for your comments. We really appreciate that you took the time and effort to critically read our protocol. We completely agree with you, that a Systematic Review (SR) should include the latest search date possible. We started this project at the beginning of November 2021. To be totally honest, we submitted this protocol in February to another journal. However, after three months of waiting time, we were informed, that they no longer publish protocols of SRs, which is why we resubmitted to PLOS One in a second step. As you might imagine, this was rather unfortunate for us, but we decided to continue to work on the SR, as we believed in the strength and accuracy of our project. By now, the final SR has progressed quite far, which is why we are rather reluctant to change the search date now. 

Second, in the Abstract and the Search strategy, they had better add Embase as one of the databases they will search. Cochrane Handbook for Systematic Reviews of Interventions Version 6.3, 2022 has recommended the search for CENTRAL, MEDLINE, and Embase in Cochrane reviews in the section 4.3.1.1 Introduction to bibliographic databases (Higgins JPT, Thomas J, Chandler J, Cumpston M, Li T, Page MJ, Welch VA (editors). Cochrane Handbook for Systematic Reviews of Interventions version 6.3 (updated February 2022). Cochrane, 2022. Available from www.training.cochrane.org/handbook.).

Thank you very much for your suggestion. We would like to answer in detail of why we did not include Embase. First, as you might have noticed, we follow the JBI methodology for systematic reviews of effectiveness evidence and not the Cochrane Handbook. In JBIs guideline Embase is listed as a potential not an obligatory source.

Second, a more practical reason: in contrast to most other sources, Embase has a restricted access. Unfortunately, we did not have access to Embase through our institution. 

To compensate for the deficit of non-existent access, we included the Cochrane Central Register of Controlled Trials (CENTRAL). Luckily, CENTRAL records are taken from Embase, which is why we believed that this information source is included at least indirectly. 

Third, in the Abstract and the Search strategy, they had better add ClinicalTrials.gov and WHO ICTRP as the databases they will search. These databases are important to search ongoing studies.

Thank you very much for your suggestion. The focus of this review was to find published studies on this topic, which is why we did not add ClinicalTrials and WHO ICTRP explicitly. We are aware that we may not be able to identify a risk of publication bias in the meta-analysis in focusing on published studies only. However, as we included CENTRAL, we include ClinialTrials and WHO ICTRP anyways, as those records are also present in CENTRAL. 

Fourth, they had better add “all adverse events” to outcomes.

Thank you for this suggestion. We added the outcome.

Fifth, they had better revise the search strategies. I consider that the standard search strategy for systematic reviews is “Participants” AND “Interventions” AND the RCT filter. The standard search strategy includes two “AND”. However, the search strategy in S1 Appendix includes four “AND” in #34 in Medline (Ovid). The search strategy may have the possibility to miss eligible studies by using too many “AND”. Therefore, I recommend they consult an information specialist, a search coordinator, or a librarian to develop search strategies for all databases to conduct a comprehensive search.

Thank you very much for this comment. We thought about your suggestion and your concerns for a long time. However, we decided against changing the search term at this stage. This is for multiple reasons. We tested and optimized the search strategy for several months. In this process, we spoke to experienced SR authors, followed search strategy guidelines 1,2 and analysed the search strategy of similar articles very closely. 

You are absolutely right that most SRs use three sets of terms: population/health condition, intervention, and study type. Amorataris and Riitano have also suggested that outcomes can be included as well. But the norm does not mean that this has to be like this all the time. As explained in the Cochrane Handbook, reviews with complex interventions can deviate. Among others, the handbook suggests to break a concept into two or more subconcepts, use iterative searches, and use citation searching on key articles. We believe that a (true) biopsychosocial intervention with a combination of physical, psychological, or social components is a rather complex intervention allowing us to deviate from the strict three-term logic.

We would like to take the time to explain our search strategy in detail. 

# 1 to 5 focuses on the Population

# 6 to 14 focuses on the physical part of the intervention.

# 26 to 32 focuses on the psychosocial part of the intervention.

# 16 to 22 restricts the search on primary care. 

# 23 to 25 restricts on study type (RCT).

Therefore, we have four 'AND' as we divide the intervention concept into two parts and restrict our search for primary care. A restriction on primary care has already been seen in similar reviews (for example: van Erp, R.M.A., Huijnen, I.P.J., Jakobs, M.L.G., Kleijnen, J. and Smeets, R.J.E.M. (2019), Effectiveness of Primary Care Interventions Using a Biopsychosocial Approach in Chronic Low Back Pain: A Systematic Review. Pain Pract, 19: 224-241. https://doi.org/10.1111/papr.12735). 

Arguably, we could have combined the physical and psychosocial part of the intervention in one term. We chose not to, to reach a reasonable precision in our results. As you might have seen, we included very broad concepts of physical and psychological types of intervention. As suggested by Cochrane, we tested our search strategy iteratively. A mere combination of the different parts resulted in a very high number of identified studies. 

From previous/similar reviews and hand-search, we identified key-articles a priori to check if they were identified by our search string. A reduction to two subconcepts still identified all key articles and reduced the number of identified studies significantly, as we included a stricter way of selection. As you described righteously, we could not be sure if we still had a comprehensive search strategy. This is why we included citation searching of key articles, as well as citation searching and forward searching of all identified articles. In doing so, we believe that we still have enough sensitivity in our search strategy without blowing up the results unreasonably. 

As mentioned above, we have rather progressed in the SR by now. Of course, a SR is only as good as the included literature. To erase remaining doubts that arose after your comment, we validated our search strategy.

We validated our strategy with the strategy of van Erp et al (as cited above). As they had a very similar research question, we used their strategy and compared the identified results with our results. As they used two “ANDs” and had a broader definition of the intervention, we expected additional search results. Indeed, as an addition to our 2.132 records after removal of duplicates, we found 411 more records on PubMed. We selected all of them and found that none of those additional articles met our scope or inclusion criteria. 

These are the reasons, why we would like to stick to the search strategy as we are certain that we were able to identify all relevant articles. 

Reviewer #2: The age range considered is from 18 years where chronic LBP might not be relevant also the exercises program are not defined well.

Dear Dr. Bhojara,

Thank you very much for taking the time and effort to read our protocol and the constructive feedback you gave us.

It is true that the prevalence of CLBP increases with age. Also, many interventions target participants between 30 and 60 years of age. However, CLPB is also seen in younger individuals (18 to 29 years). Estimates range from a prevalence of 2 to 12%3. As CLBP is prevalent in younger individuals as well, it could be possible that studies include patients from the age of 18 onwards. As we already have quite strict inclusion criteria, we did not want to restrict further based on age.

Thank you for your suggestion to explain exercises better. We updated this section in the manuscript.

---

## [Decision Letter · Decision Letter 1]

19 Aug 2022

The effectiveness of low-dosed outpatient biopsychosocial interventions compared to active physical interventions on pain and disability in adults with nonspecific chronic low back pain: a protocol for a systematic review with meta-analysis

PONE-D-22-15969R1

Dear Dr. Hochheim,

We’re pleased to inform you that your manuscript has been judged scientifically suitable for publication and will be formally accepted for publication once it meets all outstanding technical requirements.

Kind regards,

Fatih Özden, PhD

Academic Editor

PLOS ONE

Additional Editor Comments (optional):

Reviewers' comments:

Reviewer's Responses to Questions

**Comments to the Author**

1. Does the manuscript provide a valid rationale for the proposed study, with clearly identified and justified research questions?

Reviewer #1: Yes

2. Is the protocol technically sound and planned in a manner that will lead to a meaningful outcome and allow testing the stated hypotheses?

Reviewer #1: Yes

3. Is the methodology feasible and described in sufficient detail to allow the work to be replicable?

Reviewer #1: Yes

4. Have the authors described where all data underlying the findings will be made available when the study is complete?

Reviewer #1: Yes

5. Is the manuscript presented in an intelligible fashion and written in standard English?

Reviewer #1: Yes

6. Review Comments to the Author

You may also provide optional suggestions and comments to authors that they might find helpful in planning their study.

Reviewer #1: The authors have revised the manuscript thoroughly. I have no additional comments about the manuscript.

7. PLOS authors have the option to publish the peer review history of their article (what does this mean?). If published, this will include your full peer review and any attached files.

Reviewer #1: **Yes: **Masahiro Banno, MD, PhD

---

## [Editor Report · Acceptance letter]

23 Aug 2022

PONE-D-22-15969R1 

The effectiveness of low-dosed outpatient biopsychosocial interventions compared to active physical interventions on pain and disability in adults with nonspecific chronic low back pain: a protocol for a systematic review with meta-analysis 

Dear Dr. Hochheim:

I'm pleased to inform you that your manuscript has been deemed suitable for publication in PLOS ONE. Congratulations! Your manuscript is now with our production department. 

Kind regards, 

on behalf of

Dr. Fatih Özden 

Academic Editor

PLOS ONE